# Arrhythmias after COVID-19 Vaccination: Have We Left All Stones Unturned?

**DOI:** 10.3390/ijms241210405

**Published:** 2023-06-20

**Authors:** Nino Cocco, Gregor Leibundgut, Francesco Pelliccia, Valeria Cammalleri, Annunziata Nusca, Fabio Mangiacapra, Giulio Cocco, Valerio Fanale, Gian Paolo Ussia, Francesco Grigioni

**Affiliations:** 1Department of Cardiovascular Sciences, Campus Bio-Medico University of Rome, Via Álvaro del Portillo 21, 00128 Rome, Italy; 2University Heart Center, University Hospital Basel, Petersgraben 4, 4053 Basel, Switzerland; 3Department of Clinical, Internal Medicine, Anesthesiology and Cardiovascular Sciences, Sapienza University of Rome, 00186 Rome, Italy; md4151@mclink.it; 4Unit of Ultrasound in Internal Medicine, Department of Medicine and Aging Sciences, University of Chieti G d’Annunzio, 65122 Chieti, Italy

**Keywords:** SARS-CoV-2 vaccine side effect, vaccines and heart rhythm disorders, spike protein and cardiac arrhythmias, ACE system and heart conduction, angiotensin II and cardiac arrhythmias

## Abstract

SARS-CoV-2 vaccination offered the opportunity to emerge from the pandemic and, thereby, worldwide health, social, and economic disasters. However, in addition to efficacy, safety is an important issue for any vaccine. The mRNA-based vaccine platform is considered to be safe, but side effects are being reported more frequently as more and more people around the world become treated. Myopericarditis is the major, but not the only cardiovascular complication of this vaccine; hence it is important not to underestimate other side effects. We report a case series of patients affected by cardiac arrhythmias post-mRNA vaccine from our clinical practice and the literature. Reviewing the official vigilance database, we found that heart rhythm disorders after COVID vaccination are not uncommon and deserve more clinical and scientific attention. Since the COVID vaccine is the only vaccination related to this side effect, questions arose about whether these vaccines could affect heart conduction. Although the risk–benefit ratio is clearly in favor of vaccination, heart rhythm disorders are not a negligible issue, and there are red flags in the literature about the risk of post-vaccination malignant arrhythmias in some predisposed patients. In light of these findings, we reviewed the potential molecular pathways for the COVID vaccine to impact cardiac electrophysiology and cause heart rhythm disorders.

## 1. Introduction

SARS-CoV-2 vaccination has significantly reduced disease-related hospitalizations and mortality, protecting the healthcare and economic systems from an imminent crash [1]. However, in addition to efficacy, safety is an important issue for any vaccine. Since all COVID-19 vaccines are new drugs, monitoring their adverse events post-approval is essential. Some questions regarding the safety of COVID-19 vaccines have been raised due to sparse reports of severe systemic reactions after vaccination [2]. The vaccines based on the mRNA vaccine platform are considered safer and are spreading all over the world; nonetheless, side effects are being reported. Myocarditis and pericarditis are the most commonly highlighted [3,4], but there is a side effect which, in our view, has not yet been thoroughly investigated namely heart rhythm disorders. We detected post-vaccine arrhythmias in several patients in our clinical practice and further investigated the issue. A review of case reports and clinical studies on the topic and an analysis of the data from official vigilance databases of the mRNA vaccine were performed. We found that heart rhythm disorders are more common than myopericarditis, the “most famous” vaccine side effect, and deserve more clinical and scientific attention. Moreover, unlike myocarditis, none of the other vaccines were described in case reports or other publications to have this side effect.

Since less than 10% of the arrhythmias are symptomatic [5,6], epidemiological comparison of the incidence of heart rhythm disorders in patients pre- and post-COVID vaccination is not achievable from the available data. Still, the disturbance of heart rhythm is not a minor issue and, considering some red flags reported in the literature, it is very important to properly investigate the topic. In light of these findings, we analyzed the molecular pathway by which the mRNA COVID vaccine can impact cardiac electrophysiology and bring forth heart rhythm disorders.

## 2. Data from the Literature and Vigilance Databases

We experienced several patients complaining of palpitations or worsening of palpitations and arrhythmic events after mRNA COVID vaccination in our clinical practice. The review of the literature for post-vaccine heart rhythm disorders yielded several cases of heart rhythm disorders after mRNA COVID vaccination [7,8,9,10,11,12,13] (Table 1). Interestingly the COVID vaccine was the only one that was been found to be related to this side effect.

The analysis of the vigilance database showed that one of the most frequent symptoms described after SARS-CoV-2 vaccination is palpitations [13,14]. Tachycardia or increased heart rate can be a physiological or stress-related response after vaccination and is also called an immunization stress-related response (ISRR) [15,16]. This component in a patient that has recently undergone a new vaccine administration might have played an important role and has to be taken into account; however, there are some important red flags related to vaccine-induced arrhythmias that we cannot overlook.

**Table 1 ijms-24-10405-t001:** Case reports regarding cardiac arrhythmias after COVID vaccination.

Authors	Title	Year	N. of Cases	Centre/Nation	Type of Vaccine	Type of Arrhythmia
Aiba T et al. [5]	“Frequent Premature Ventricular Contraction and Non-Sustained Ventricular Tachycardia After the SARS-CoV-2 Vaccination in Patient with Implantable Cardioverter Defibrillator Due to Acquired Long-QT Syndrome”	June 2021	1	Japan	Pfizer-BioNTech	Premature ventricular contraction and non-sustained polymorphic ventricular tachycardia
Abrich and Olshansky [6]	“Torsades de pointes following vaccination for COVID-19”	June 2022	1	USA	Pfizer-BioNTech	Torsades de pointes-ventricular fibrillation
Beccarino et al. [7]	“COVID-19 vaccination associated right ventricular outflow tract ventricular tachycardia in a structurally normal heart”	March 2022	1	USA	Pfizer-BioNTech	Sustained unstable ventricular tachycardia
Slater et al. [8]	“VT storm in long QT resulting from COVID-19 vaccine allergy treated with epinephrine”	November 2021	1	Canada	Moderna	Ventricular tachycardia storm in long QT
M. Teresa Marco García et al. [9]	“Tachycardia as an undescribed adverse effect to the Comirnaty© vaccine (BNT162b2 Pfizer-BioNTech COVID-19 vaccine): Description of 3 cases with a history of SARS-CoV-2 disease”	May 2022	3	Spain	Pfizer-BioNTech	Ventricular Tachycardia and extrasystoles in physicians
Hoffer Etienne et al. [10]	“Transient but recurrent complete heart block in a patient after COVID-19 vaccination—A case report”	April 2022	1	Belgium	Pfizer-BioNTech	Transient but recurrent complete heart block
Kyung Hee Lim and Jong-Sung Park [17]	COVID-19 Vaccination-Induced Ventricular Fibrillation in an Afebrile Patient with Brugada Syndrome	April 2022	1	Korea	Pfizer-BioNTech	Ventricular fibrillation

In August 2021, Lehmann et al. focused on the anonymously reported adverse events of vaccinated healthy individuals in publicly available databases or reports as well as data retrieved from web reports of EudraVigilance of the European Medicines Agency (EMA) and the safety report from the German PEI” [11]. The findings highlighted alarming cases of thrombotic/embolic events related to Vaxzevria (Astra Zeneca) but also some cardiac complications related to Comirnaty (Pfizer mRNA vaccine), where the cardiac arrhythmias ranked second after hypertensive crisis: “because of their frequency and importance, they should exert a signaling function and need to be better communicated” [18].

Patone M. et al. reported an increased risk of cardiac arrhythmia at 1–7 days following a second dose of mRNA-1273 (Moderna) [19]. Norazida Ab Rahman et al., in an analysis of a population-based study covering over 20 million vaccinated individuals in Malaysia, reported the risk of post-vaccination hospitalization for predefined diagnoses among patients vaccinated with BNT162b2, CoronaVac, and ChAdOx1 vaccines. This self-controlled case-series study found a significant association between COVID-19 vaccination and cardiac arrhythmia, “although no study has found an association between arrhythmia and COVID-19 vaccines, our findings suggest that arrhythmia following vaccination in our population is an adverse event of special interest (AESI) of concern” [20]. Chupong Ittiwut et al. highlighted that the presence of proarrhythmic genetic mutation increases the risk of life-threatening arrhythmias in post-COVID vaccine patients. In particular, SCN5A variants could be associated with sudden death within 7 days of COVID-19 vaccination, regardless of vaccine type, number of vaccine doses, and presence of underlying diseases or postvaccine fever. Because of these observations, the authors closely monitored the individuals that harbor variants in SCN5A, and possibly in other genes that predispose to cardiac arrhythmias or cardiomyopathies, for 7 days after the administration of COVID-19 vaccines, regardless of preexisting underlying diseases and the presence of vaccination-associated fever [21].

These data add to other ambiguous red flags previously reported in the literature, such as the study from Sun CLF that reported the trends of calls for sudden cardiac arrest (defined as electrical malfunction) increased in the vaccination period in the younger population [22], and from Maffetone et al. regarding the anomalous number of sudden deaths in athletes “post-COVID-19 and/or vaccination worldwide” with a 500% increase only in the FIFA members [23].

The analysis of the recent data of the UK Yellow Card Scheme until 29 June 2022, the mRNA Pfizer/BioNTech vaccine analysis print confirmed that palpitations (6174, 3.6%) are one of the most frequently reported adverse events, with a frequency comparable with rash (6745) [14]. Furthermore, 3610 (2%) cases of rate and rhythm disorders were reported. Myocarditis, the “star” of the vaccine-related complications, had 777 (0.45%) reports, and including pericarditis, the number of reports went up to 1310 (0.77%), which is far lower in comparison to the heart rhythm disorder. Looking at the entire population, myocarditis was reported in 3 per 100,000 patients and heart rhythm disorders in 14.4 per 100,000 after the Pfizer vaccine [14]. 

The trend from the COVID-19 Moderna Vaccine Analysis Print is not much different [24]. Ajmera KM et al. analyzed the Vaccine Adverse Event Reporting System (VAERS) data in the USA and reported a suspect incidence of atrial fibirilation: more than 2000 AF post-COVID-19 vaccination [25]. These data brought Ayeesha Kattubadi et al. to investigate the topic, and their pharmacovigilance analysis also confirmed the correlation between the Pfizer COVID vaccine and atrial fibrillation [26].

To ascertain the relationship between vaccination and arrhythmic events, Naruepat Sangpornsuk et al. studied the occurrence of arrhythmia in patients with cardiac implantable electronic devices before and after a SARS-CoV-2 vaccination and reported that the incidence of arrhythmia was significantly increased after the SARS-CoV-2 vaccination with a 73% increase in the incidence rate of sopraventricular arrhythmias and a 316% increase in the incidence ratio of ventricular arrhythmias [27].

Myocarditis is a described side effect of the vaccination and was also reported in the context of other vaccines, like the influenza vaccine, but heart rhythm disorders were never associated with vaccination other than the COVID vaccines [28]. The influenza vaccination represents an ideal control for the ongoing COVID-19 vaccination because, on the one hand, seasonal influenza viruses share with coronaviruses substantial similarities regarding symptomatology, infectivity, pathogenicity, lethality, and transmission, and, on the other hand, a large proportion of the adult populations in the EU and US is vaccinated against influenza every season. Therefore, Diego Montano estimated the absolute and relative risks of serious adverse reactions associated with the COVID-19 vaccine reports in comparison to influenza vaccines drawing on data from the VAERS and EudraVigilance databases [2]. “The rationale behind the comparison of adverse reaction risks between COVID-19 and influenza vaccines allows us to assess the potential risk profile of the new vaccines. Thus, COVID-19 vaccines and influenza vaccines were compared on the common metric of the probability of observing serious adverse reactions following vaccination”. A higher risk of reporting serious adverse reactions was observed for the COVID-19 vaccines in comparison to the influenza vaccines from the European and United States pharmacovigilance systems, and the cardiac arrhythmias were between the adverse reaction with the largest relative risk (RR 99% CI of 83.10 [43.59–158.41]) [2].

Ao Shi et al., in a recent meta-analysis, explored the incidence of arrhythmia after COVID-19 vaccination and compared it with the incidence of arrhythmia after non-COVID-19 vaccination. The overall incidence of arrhythmia from 36 studies (1,528,459,662 vaccine doses) was 291.8 (95% CI 111.6–762.7) cases per million doses. The incidence of arrhythmia was significantly higher after COVID-19 vaccination (2263.4 cases per million doses) than after non-COVID-19 vaccination (9.9 cases per million doses). Compared with COVID-19 vaccines, influenza, pertussis, human papillomavirus, and acellular pertussis vaccines were associated with a significantly lower incidence of arrhythmia. The incidence of tachyarrhythmia was significantly higher after COVID-19 vaccination (4367.5 cases per million doses) than after non-COVID-19 vaccination (25.8 cases per million doses). Arrhythmia was also more frequent after the third dose of the COVID-19 vaccine (19,064.3 cases per million doses) than after the first dose (3450.9 cases per million doses) or second dose (2262.5 cases per million doses) [29].

Possible interference of COVID-19 vaccination on cardiac ion channels was suspected as a result of some clinical case reports. There has been a described mRNACOVID vaccination unmasking type-1 Brugada ECG pattern not related to body temperature, and concern that vaccination may induce life-threatening ventricular arrhythmias in patients with Brugada syndrome [17,30]. Kyung Hee Lim et al. described a case of a 43-year-old man with no history of spontaneous type-1 Brugada ECG pattern, no history of faintness or syncope or familial history of sudden cardiac death, who presented with cardiac arrest 2 days after the second coronavirus disease 2019 (COVID-19) vaccination with an mRNA vaccine. Electrocardiograms showed ventricular fibrillation and type-1 Brugada pattern ST segment elevation. The patient reported having no symptoms, including febrile sensation. There were no known underlying cardiac diseases to explain such electrocardiographic abnormalities. ST segment elevation completely disappeared in two weeks. Although there were no genetic mutations or personal or family history typical of Brugada syndrome, flecainide administration-induced type-1 Brugada pattern ST segment elevation. “This case suggests that COVID-19 vaccination may induce cardiac ion channel dysfunction and cause life-threatening ventricular arrhythmias in specific patients with Brugada syndrome” [31]. These observations are in line with the results of Chupong Ittiwut et al.’s study [21], in which SCN5A, a variant associated with Brugada Syndrome, was related to a major risk of life-threatening arrhythmias.

In a recent systematic review and metanalyses focusing on the risks of cardiac arrhythmia associated with COVID-19 vaccination, Mohammed H. Abutaleb et al. highlighted that mRNA vaccines were associated with a higher risk of arrhythmia compared to vector-based vaccines; in the same study, inactivated vaccines showed the lowest IR of arrhythmia [32]. “Studies have reported that various cardiac arrhythmias complicate COVID-19 vaccination, although the incidence of cardiac arrest is low. Myocardial inflammation, cytokine storm, autonomic dysfunction, and ion channel dysfunction appear to be associated with the occurrence of COVID-19 vaccination-induced cardiac arrhythmias; however, the exact pathophysiologic mechanisms responsible remain unknown” [33,34].

The question remains if the mRNA COVID vaccine has a pathophysiologic impact on heart electrophysiology that could explain an effect on heart rhythm.

## 3. The Molecular Pathway behind the Heart Electrophysiology Alteration

The most relevant cardiovascular side effect of the vaccines is considered to be myocarditis [35,36]. Vaccine-induced myopericarditis is a significant player in vaccine cardiovascular complications, and many experts consider heart rhythm disorder a consequence of vaccine-induced myocarditis [35,36,37,38]. Incidents of myocardial inflammation causing heart rhythm disorders has solid and strong literature; however, the signs of inflammation were not detected in the cases reported, and we want to highlight other molecular pathways that could be involved in the pathophysiology of vaccine-induced cardiac arrhythmia.

The target of the COVID-9 vaccine is the spike protein, which is the protein interfering with the Angiotensin-converting enzyme 2 (ACE2) receptor representing the door of entry of the virus during the infection. ACE2 is not only a receptor but also an important catalytic site, and its activity represents a pillar for the balance of the Renin-Angiotensin system (RAS), one of the most important systems for cardiovascular homeostasis preservation. Indeed, the RAS depends on the tissue angiotensin system (ACEs) that is the product of a balance between Angiotensin II (AT II), derived by the enzymatic activity of ACE from Angiotensinogen, and Angiotensin 1/7 and 1/9, derived by the activity of ACE2 from AT II. AT II has pro-hypertensive, prothrombotic, pro-inflammatory and pro-fibrotic properties and Angiotensin 1–7 anti-hypertensive, anti-thrombotic anti-inflammatory and anti-fibrotic pathways. Thus, ACE2 has a fundamental role in local and systemic hemodynamics, lower blood pressure, promotes vasodilation and has antioxidant and antiproliferative effects, promoting the production of Angiotensin 1–7 from AT II. ACE2 is highly expressed in the heart, more than in other organs such as lungs or kidneys and is secreted into the plasma [39,40,41]. It was demonstrated that the spike protein of SARS-CoV-2 competes with AT II for ACE2 in terms of internalization, and the binding blocks’ ACE2 activity and expression [41] favoring inflammation, thrombosis and endothelial damage due to AT II accumulation [42,43,44]. The alteration of ACE is considered the pathophysiological key to COVID-related cardiovascular complications [42,43,44,45,46].

Vaccine-induced spike protein has a high affinity for ACE2 and can promote ACE2 internalization and degradation, bringing ACE imbalance like in COVID infection. This could explain some described “COVID like” side effects, an effect that Angeli et al. called “the spike effect” [47]. However, what is the role of ACE in heart electrophysiology?

ACE is also a strong regulator of heart conduction and greatly impacts heart electrophysiology. COVID-19 is an infectious disease characterized by pneumonia, but complications affecting the cardiovascular system are the most serious and fatal [42]. Myocarditis, thromboembolic events, and hypertensive emergencies, as well as arrhythmias, are frequent [43] and are the second most common cardiac complications [48,49,50]. Pathophysiology of arrhythmias involved inflammation and hypoxia, but a direct interaction with the spike protein and cardiac conduction system was found in patients who died from arrhythmias. The immunohistochemical analysis detected the presence of the spike and nucleocapsid protein in the conduction system of a patient who died from heart rhythm disorders [51]. Moreover, heart rhythm disorders have been reported in patients not presenting with severe inflammatory responses [44,52].

Like in other cardiovascular complications, spike protein interfering with ACE2 and unbalancing ACE might be the key to understanding the occurrence of SARS-CoV-2 and vaccine-related heart rhythm disorders [53,54]. The cardiac conduction system has a high density of ACE receptors [46]. The effect of AT-II has been under investigation for a long time. In 1981, Kass and Blair demonstrated that nanomolar concentrations of AT II have marked effects on the electrical and mechanical activity of the Purkinje system [55].

Angio 2 induced changes in membrane currents and increased the height and duration of the plateau phase of the action potential in isolated cardiac Purkinje fibers of a calf heart [55]. Saito et al. demonstrated that there are Angio II binding sites highly localized in the conduction system of the heart, using autoradiography and computerized microdensitometry of the tissue section of the rat heart, which could influence the chronotropy [56]. De Mello and coworkers described the role of Angio II in arrhythmogenesis. They identified the capacity of Angio II to reduce the refractoriness and the resting potential of the muscle trabeculae isolated from an adult rat ventricle acting on specific Angio II receptors [57]. In addition, they reported Angio II to reduce coupling at the gap junctions altering the junctional conductance and favoring electrical uncoupling in isolated ventricular cell pairs of adult rats [57,58].

These data support the hypothesis that the peptide has deleterious effect on heart conduction favoring reentrant rhythms and abnormal automaticity [59]. Reentry is the cause of the most lethal tachyarrhythmias and is caused by a decrease in conduction velocity and/or a reduction in refractory period [57]. In a porcine model of myocardial infarction, AT II infusion reduced ventricular refractoriness favoring spontaneous ventricular arrhythmias [60]. Different studies have also investigated the molecular mechanisms underlying Ang II-related arrhythmias in ventricular myocytes [61,62]. Zankov et al. demonstrated that Angio II can act on the delayed rectifier K current (IK), the major repolarizing outward currents of action potentials [63]. Gunasegaram et al. described the regulation of sarcolemmal Na (+)/H (+) exchanger activity by angiotensin II in adult rat ventricular myocytes and the negative actions of AT1 and AT2 receptors in this context [64]. Wagner et al. showed that Ang II increase diastolic Ca leak from the sarcoplasmic reticulum and increases late Ca and sodium currents favoring Ca channel re-opening and early afterdepolarizations [65], demonstrating that Ang II is also important for cellular conductance and coupling. Finally, Sotoodehnia et al. described the association between genetic variation in ACE-related pathways with sudden cardiac death “that support the need for further investigation of ACE-related pathways in arrhythmogenesis” [66]. These data confirmed that ACE system balance is crucial in cardiac electrophysiology and conductance homeostasis, influencing all the steps of cardiac conduction, from depolarization to cellular coupling and repolarization (Figure 1).

We described the strong impact of the ace system on cardiac conduction, but there are some points that must be clarified. (1) If the pathophysiology is Angio 2-related, why are patients with a baseline deficiency of ACE2 receptor activity (advanced age, cardiovascular risk factors, and previous cardiovascular events) significantly affected by SARS-CoV-2 infections but not vaccination side effects? Indeed, younger patients seem to be more involved. (2) Why do vector-based vaccines and mRNA-based vaccines share the same target protein but present different complications? And (3) how can we explain longstanding arrhythmias and/or tardive arrhythmias, especially alongside arrhythmias that tend to be the worst after the second or third dose?

Angeli et al. highlighted that ACE2s are not the exclusive angiotensinases in nature [47]. Other angiotensinases involved in the processing of Ang II to Ang1,7 may exert a counterbalance influence in the detrimental interactions between Spike proteins and ACE2 receptors. Specifically, the AngioII-Angio1,7 axis of the RAS encompasses three enzymes (carboxypeptidases) that form Angio1,7 directly from (by cleavage) Angio II: ACE2, prolyl carboxypeptidases (PRCP), and prolyl oligopeptidase (POP). PRCP has protective effects that cleave the C-terminal amino acid of Ang II to form Ang1,7. However, the increased catalytic activity of POP and PRCP is not typical in the young but more pronounced in elderly individuals with comorbidities or previous cardiovascular events [67]. Thus, the adverse reactions to COVID-19 vaccination associated with Angio II accumulation in younger subjects could be related to this aspect. Important to remark that the activity of these two carboxypeptidases and neprilysin pathways (other pathways described) remains substantially unchanged during the acute phase of COVID-19 infection. Hence, the plasma concentration of Angio 1–7 remains stable in COVID-19 patients, and it appears insufficient to prevent the harmful effects of Ang II. This could explain the increased risk of severe COVID-19 among patients with specific phenotypes of ACE2 deficiency and could explain the paradox of the vaccine-protective/COVID-unprotective effect [67,68].

Adenovirus-vector and mRNA vaccines promote substantially different innate responses that will certainly influence the nature of characteristics of immune responses and possible adverse reactions Oxford-AstraZeneca vaccine might induce higher levels of specific T cells, whereas mRNA vaccines might induce higher antibody titers [69,70].

One way to explain the long-term side effect of the COVID disease and the COVID vaccine is through the lens of antibiotype immune response. Every antibody that is induced and specific for an antigen (termed “Ab1” antibody) has immunogenic regions capable of inducing specific antibodies against Ab1 antibodies (Ab2 antibody). These Ab2 antibodies specific for Ab1 can structurally resemble that of the original antigens [71,72]. As a result of this mimicry, Ab2 antibodies also have the potential to bind the same receptor that the original antigen was targeting, mediating profound effects on the cell that could result in pathologic changes, particularly in the long term. Ab2 alone can even mimic the deleterious effects of the virus particle itself [70,71,72]. Myocarditis after vaccine administration bears striking similarities to the myocarditis associated with Ab2 antibodies induced after some viral infections [73]. Antibodies could also mediate long-standing cardiovascular effects and heart rhythm disorders of SARS-CoV-2 infection or vaccines, given the expression of ACE2 in the heart [71,74,75,76]. The presence of antibodies specific for ACE2 in patients was demonstrated with COVID or a history of infection and was hypnotized to result from anti-idiotypic antibodies to the spike protein [75]. Autoantibodies against ACE2 increase the severity of COVID-19, lowering the activity of soluble ACE2 in plasma after the infection, which can present poor outcomes, and was speculated the potential implication of autoantibodies against ACE2 also in post-vaccination complications [75,76].

Donoghue et al. showed that also ACE2 overexpression could cause conduction disturbances and lethal ventricular arrhythmias due to a connexin dysregulation, gap junction remodeling and profound electrophysiologic disturbances [77]. Therefore, ACE2 protection works in a context of balance where the overactivity of ACE2 could also be unfavorable [78,79].

Gitelman and Bartter syndromes (GS/BS) are rare genetic tubulopathies that have endogenously increased levels of ACE2 and were demonstrated that these diseases are characterized by microvascular dysfunction and, in particular, by an increased risk of cardiac arrhythmias that could not only be justified by the electrolyte’s alteration [80,81]. These syndromes offer a good opportunity to investigate the pathophysiology of COVID complications. Interestingly, Caló et al. demonstrated these genetic disorders to be protective against COVID infection and complication, confirming the protective role of ACE 2 in COVID and the ambiguous role in the pathophysiology of the arrhythmias [82].

In conclusion, spike proteins from COVID or vaccine bind with high-affinity ACE2 could potentially cause an imbalance in ACEs, which plays a role in COVID and vaccine complications. Heart rhythm conduction is significantly affected by ACEs, and its alteration could provoke heart rhythm disorders. Cardiac arrhythmias are not a minor issue and must be properly investigated. Our patients were treated with metoprolol because of its demonstrated efficacies on arrhythmias and its effect on the Angio II [83,84]. The treatment was very effective.

The spike protein—whether from COVID or vaccine—could represent the key to understanding the molecular pathway of the heart conduction abnormality. Binding with high affinity to ACE2 could potentially result in an ACE imbalance, thus promoting cardiac complications [85,86]. Heart rhythm conduction is highly affected by ACEs, and its alteration may promote heart rhythm disorders (Figure 1). Cardiac arrhythmias have a high mortality and morbidity and need further investigation. Metoprolol has a beneficial effect on the AT II pathway [85,86], and its use as a treatment was very effective.

## 4. Conclusions

We treated several patients at our clinical practice complaining of palpitations or worsening of palpitations after an mRNA COVID vaccination. Concordantly, data from the vigilance database and the literature report cases of heart rhythm disorders in the post-mRNA vaccination period and malignant arrhythmias in predisposed patients, suggesting a potential impact on the cardiac conduction system. Otherwise, we have not encountered heart rhythm disorders after administering other vaccines and found no signs of arrhythmias in the vigilance databases. Since cardiac arrhythmias have high morbidity and mortality, and considering the red flags in the literature describing the risk of malignant arrhythmias in patients with genetic arrhythmic predisposition [19], the topic must be further investigated [70].

We analyzed the molecular pathways of new mRNA vaccines and found they may potentially provoke electric disturbances, specifically regarding their effects on the ACE system and the role of the spike protein in the contest of cardiovascular complication [87,88]. In addition, these observations raise concerns regarding the central role of the ACE system on heart conduction homeostasis and possible detrimental impact of iatrogenic interference with this system in this context.

Finally, it is important to note that in terms of the risk–benefit ratio of COVID-19 and our findings, the use of vaccinations is clearly favored. However, we should consider a thorough risk–benefit assessment in specific populations.

## Figures and Tables

**Figure 1 ijms-24-10405-f001:**
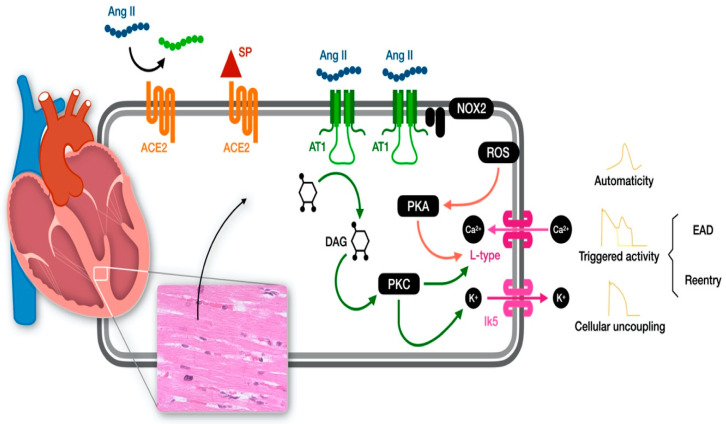
Molecular pathways of the impact of Angiotensin II on cardiac automaticity, trigger activity and cellular uncoupling.

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
