# Peer review of "Arrhythmias after COVID-19 Vaccination: Have We Left All Stones Unturned?"

_ijms, 2023, doi:10.3390/ijms241210405_

Round 1

Reviewer 1 Report

Several minor suggestions:

1.       In the abstract and Conclusion, be careful to use ‘we investigated…’. In this review paper, you don’t do any experiments to investigate the cardiac rhythm disorders–related mechanisms. You actually overviewed potential molecular pathways from other publications. I suggest using ‘reviewed’ or ‘list’ instead of ‘investigated’.

2.       In introduction, ‘SARS-CoV-2 vaccination has significantly reduced disease-related …’ needs references. ‘Some questions regarding the safety of COVID-19 vaccines…’ needs references regarding spars reports. ‘In light of these findings, we hypothesize on the mechanism of the impact …’ inappropriately used the word ‘hypothesize’.  What is your hypothesis? How do you demonstrate your hypothesis? Please re-write this sentence to make it clear.

3.       The molecular pathways associated with heart rhythm disorders after the vaccine are discussed. Could you highlight and describe cell or animal models used in other publications to help readers understand in which models these potential mechanisms are discussed?

Author Response

Dear Editor

Thank you for your comments concerning our manuscript entitled “Arrhythmias after Covid-19 vaccination: have we left all stones unturned?”

1. The comments are all valuable and very helpful for revising and improving our paper, as well as the important guiding significance to our research. We have studied the comments carefully and have made corrections which we hope to meet with approval. The main corrections in the paper and the replies to the reviewer’s comments are as follows

  1. In the abstract and Conclusion, be careful to use ‘we investigated…’. In this review paper, you don’t do any experiments to investigate the cardiac rhythm disorders–related mechanisms. You actually overviewed potential molecular pathways from other publications. I suggest using ‘reviewed’ or ‘list’ instead of ‘investigated’.

Thanks for your valuable comment, we changed the word as you suggested

  1. In introduction, ‘SARS-CoV-2 vaccination has significantly reduced disease-related …’ needs references. ‘Some questions regarding the safety of COVID-19 vaccines…’ needs references regarding spars reports. ‘In light of these findings, we hypothesize on the mechanism of the impact …’ inappropriately used the word ‘hypothesize’.  What is your hypothesis? How do you demonstrate your hypothesis? Please re-write this sentence to make it clear.

Thanks for your comments, we added the references and re-write the sentence as you advised

  1. The molecular pathways associated with heart rhythm disorders after the vaccine are discussed. Could you highlight and describe cell or animal models used in other publications to help readers understand in which models these potential mechanisms are discussed?

Thanks for your comment, we specified more precisely cell and animal models used in the publications

Reviewer 2 Report

Major comments

Chapter 3 is very chaotic. A table and a figure summarizing the main molecular mechanisms are required.

Authors should merge the brief discussion and conclusion sections, eliminating the repetitions.

Minor comments

Line 125: “infulenza” not “influenca”

Line 194: “COVID-19” not “Covid”. Please correct in other point of the text

Line 269: the authors start a numbered list of question, but number 1 is missing.

Author Response

Dear Editor

Thank you for your comments concerning our manuscript entitled “Arrhythmias after Covid-19 vaccination: have we left all stones unturned?”

The comments are all valuable and very helpful for revising and improving our paper, as well as the important guiding significance to our research. We have studied the comments carefully and have made corrections which we hope to meet with approval. The main corrections in the paper and the replies to the reviewer’s comments are as follows

Major comments

  1. Chapter 3 is very chaotic. A table and a figure summarizing the main molecular mechanisms are required.

Thanks for your valuable comment. You are completely right, the pathophysiological pathways are complex and somehow could appear chaotic. A picture would have been important to summarize the molecular mechanism and clarify some steps. Actually, we have a summary picture but the journal does not accept unpublished material. We will try to ask for the chance to incorporate this picture in the paper because is very important for the understanding of the paper. You will find a picture like "Figure 1" in the reviewed paper now

  1. Authors should merge the brief discussion and conclusion sections, eliminating the repetitions.

Special thanks for your valuable comment, we merged the discussion and conclusion sections, eliminating the repetitions

Minor comments

  1. Line 125: “infulenza” not “influenca”
  2. Line 194: “COVID-19” not “Covid”. Please correct in other point of the text
  3. Line 269: the authors start a numbered list of question, but number 1 is missing.

Thanks for your suggestions